# The Neuroprotective and Antioxidant Effects of Nanocurcumin Oral Suspension against Lipopolysaccharide-Induced Cortical Neurotoxicity in Rats

**DOI:** 10.3390/biomedicines10123087

**Published:** 2022-12-01

**Authors:** Adham Salah, Mokhtar Yousef, Maher Kamel, Ahmed Hussein

**Affiliations:** 1Department of Biotechnology, Institute of Graduate Studies and Research, Alexandria University, Alexandria 5422023, Egypt; 2Department of Environmental Studies, Institute of Graduate Studies and Research, Alexandria University, Alexandria 5422023, Egypt; 3Biochemistry Department, Medical Research Institute, Alexandria University, Alexandria 5422031, Egypt

**Keywords:** lipopolysaccharides (LPS), nanocurcumin, mitochondrial biogenesis, oxidative stress, neuroinflammation, neurotoxicity, neurotransmitters

## Abstract

Lipopolysaccharide (LPS) proved to be an important tool, not only in the induction of neuroinflammatory models, but also in demonstrating the behavioral and cognitive consequences of endotoxemia. Curcumin, in its native form, has proven to be a worthy candidate for further development as it protects the dopaminergic neurons against LPS-induced neurotoxicity. However, it remains hindered by its poor bioavailability. In this study we aim to explore the possible molecular mechanism of LPS-induced neurotoxicity and the possible protective effects of orally supplemented nanocurcumin. Thirty-six adult male Wistar rats weighing 170–175 g were divided into six groups and treated with single I.P. (intra-peritoneal) dose of LPS (sigma and extracted; separately) (5 mg/kg BW) plus daily oral nanocurcumin (15 mg/kg BW). The rats were followed for 7 days after the LPS injection and nanocurcumin supplementations daily via oral gavage. After scarification, the levels of neurotransmitters, antioxidants, and amyloidogenesis markers were assessed in brain tissues. Nanocurcumin showed adequate antioxidant and neuroprotective effects, rescuing the rats which had been injected intraperitoneally with LPS endotoxin.

## 1. Introduction

Lipopolysaccharide (LPS) is the most common glycolipid on the cell surface of most Gram-negative bacteria, accounting for up to 80% of *E. coli* and *Salmonella* sp. outer membranes [1]. It is present on the outer leaflet of the outer membrane of Gram-negative bacteria, contributing to the barrier function and structural integrity, preventing passive diffusion of hydrophobic molecules such as antibiotics into the cell, and resulting in the high resistance of Gram-negative bacteria to harsh environmental conditions and antibiotics [2]. It is undeniably a source of inflammatory reactions in the brain and a possible source of many of the neurodegenerative conditions [3].

LPS proved to be an important tool not only in the induction of neuroinflammatory models [4] but also in demonstrating behavioral and cognitive consequences of endotoxemia [5], bearing in mind that LPS effects may be altered depending on the route of administration, number of injections, time of exposure, and serotype [6].

The amphipathic nature of LPS proved to be an obstacle in its extraction from Gram-negative bacteria [7]. Many isolation, purification, and characterization protocols are available, including aqueous butyl alcohol, cold ethanol, triton/Mg^+2^, trichloroacetic acid extraction at 4C, and extraction in water at 100C. For rough LPS purification with chloroform, petroleum ether, phenol, and methanol may be applied [8]. For smooth and rough LPS extraction, the trireagent method could be used, as it has proven to be successful and provides a high amount of yield [7]. Westphal’s method is still considered the gold standard for LPS extraction, as it provides a high amount of yield and can be combined with size exclusion chromatography for successful purification of LPS from the outer membrane complex of *Escherichia coli* Gram-negative bacteria [8].

LPS induces its neuroinflammatory response in mammals by binding to toll-like receptor 4 (TLR-4), which is a pattern recognition receptor (PRR) that is a critical immune response driver to bacterial infections [9]. Interleukin (IL)-6 and tumor necrosis factor (TNF)-α are also markers that indicate such inflammation induced by LPS [10], leading to the increased production of reactive oxygen species (ROS) that result in extensive and irreversible neuronal damage [11]. Despite the efforts done in many studies to provide a rescue for such damage, more investigation is needed to discover more molecules that can prevent or rescue such neurological toxicity.

Curcumin, in its native form, has proven to be a worthy candidate for further development, as it protects the dopaminergic neurons against LPS-induced neurotoxicity [12]. Hindered by its low bioavailability, the full potential of curcumin cannot be demonstrated, as only its ingestion will not lead to a noticeable effect; it will not reach the bloodstream in a reasonable concentration to provide its full protective and antioxidant benefits [13].

The challenges associated with water-insoluble molecules have been resolved using nanotechnology [14]. The nanoscale synthesis of hydrophobic pharmaceuticals improves their solubility and bioavailability [15]. Furthermore, nanosized medications can freely enter cells and target specific cytosolic regions, such as nucleic acids, proteins, and other small-sized molecules [14].

Curcumin nanoformulations have dramatically transformed how diseases are treated by improving bioavailability, cellular absorption, and permeability with increased plasma concentration [16]. For the synthesis of nanocurcumin, a variety of approaches have been established by different researchers, each showing some pros and cons. The Fessi method, nanoprecipitation, microemulsion, spray-drying, single emulsion, emulsion polymerization, solvent evaporation, antisolvent precipitation, ultrasonication, coacervation technique, ionic gelation, wet milling, solid dispersion, and thin-film hydration are the most frequent and recognized procedures [17].

Utilizing the tools provided by the field of nanotechnology to deliver nanocurcumin particles instead of the native curcumin provided a better solubility and bioavailability. It also proved to be a more effective alternative for native curcumin and it has been reported that the nanoform of curcumin is the better form for preventing neurodegenerative changes in experimentally induced cerebral malaria [15]. Nanocurcumin can cross the blood–brain barrier (BBB) to enter brain tissue, where it concentrates chiefly in the hippocampus and for a significantly prolonged retention time in the cerebral cortex (increased by 96%) and hippocampus (increased by 83%) [18].

This study aims to explore the possible molecular mechanism of LPS-induced neurotoxicity and the possible protective effects of orally supplemented nanocurcumin suspension. This work was accomplished through the assessment of cerebral cortex and hippocampal changes at different levels, including histological, neurotransmitter, amyloidogenic pathway, redox and antioxidant status, inflammatory status, and gene expression of the mitochondrial transcription factor (mTFA).

## 2. Materials and Methods

### 2.1. Materials

Sigma lipopolysaccharides (LPSs) were purchased from Sigma Aldrich, Cedex, France (Product name Lipopolysaccharides from *Escherichia coli* O111:B4, Product Number L2630, Sigma-Aldrich Chemie GmbH Eschenstrasse 5 D-82024 TAUFKIRCHEN company). The extracted LPS was obtained from *E. coli* ATCC 25922 bacteria according to El-Moslemany et al. [19]. Nanocurcumin (NT-Cur-NPs; 4020) was purchased from Nanotech, Giza, Egypt for photo-electronics, Egypt. Both LPSs were dissolved in normal isotonic saline and I.P. injected in doses of 5 mg/kg BW according to Nezić et al. [20]. The dose of nanocurcumin was 15 mg/kg BW according to Yadav et al. [21]. An amount of 70 mg of nanocurcumin were suspended in 70 mL distilled water. All reagents and chemicals were of analytical grade, and water used throughout the entire work was distilled water.

### 2.2. Extraction of LPS

Lipopolysaccharide was extracted by hot phenol–water method, as described previously, with some modifications [19]. In brief, bacterial suspensions *E. coli* ATCC 25922 (120 colony-forming units/mL) were centrifuged at 9000× *g* for 8 min. The pellets were washed twice in PBS (pH = 7.2, 0.1 M) containing 0.15 mM CaCl_2_ and 0.5 mM MgCl_2_. Bacterial pellets were resuspended in 5 mL PBS and sonicated for 10 min on ice. To eliminate contaminating proteins and nucleic acids, treatment with proteinase K, DNase, and RNase was performed. Proteinase K (100 μg/mL) (Roche, Mannheim, Germany) was added to the bacterial suspensions and the tubes were kept at 65 °C for one hour. The mixture was subsequently treated with RNase (40 μg/mL) (Roche, Mannheim, Germany) and DNase (20 μg/mL) (Roche, Mannheim, Germany) in the presence of 1 μL/mL 20% MgSO_4_ and 4 μL/mL chloroform. Incubation continued at 37 °C overnight. At the next step, an equal volume of hot (65–70 °C) 90% phenol was added to the mixtures, followed by shaking at 65–70 °C for 15 min. Suspensions were then cooled on ice, transferred to 1.5 mL tubes, and centrifuged at 9000× *g* for 15 min. Supernatants were transferred to 15 mL conical centrifuge tubes and phenol phases were re-extracted by 300 μL distilled water. Sodium acetate at 0.5 M final concentration and 10 volumes of 95% ethanol were added to the extracts. Samples were stored at −40 °C overnight in order to precipitate LPS. Tubes were then centrifuged at 4000× *g* 4 °C for 15 min and the pellets were suspended in 1 mL distilled water. Extensive dialysis against double-distilled water at 4 °C was carried out at the next step until the residual phenol in the aqueous phases was totally eliminated. Final purified LPS product was lyophilized and stored at 4 °C.

### 2.3. Nanocurcumin

As per the company of origin (Nanotech Egypt for photo-electronics) of the purchased nanocurcumin, the nanocurcumin powder was prepared by emulsification–solvent evaporation using Annaraj et al.’s method [22]. Bulk curcumin was dissolved in chloroform in the ratio of 1:2 (solution A) and PVA was dissolved in distilled water (solution B). Drop-wise addition of Solution A to boiling water at 70 °C under stirring and continued stirring for a further one hour. At the end of the reaction, an orange-colored fluffy mass was obtained. Solution B was added into solution A and the temperature was reduced below 40 °C. Stirring continued for another 2 h. The solution was allowed to settle and was stored at below −14 °C for one day. The supernatant was discarded after centrifugation and the reaction solution was dried in a water bath at 60 °C. A fine powder of PVA-loaded nanocurcumin was obtained.

PVA-loaded nanocurcumin thus obtained was characterized by UV-Vis absorption spectra that was obtained on an Ocean Optics USB2000+VIS-NIR Fiber optics spectrophotometer. Furthermore, to check the shape and size, TEM was performed on JEOL JEM-2100 (JOEL, Tokyo, Japan) high-resolution transmission electron microscope at an accelerating voltage of 200 kV. The obtained nanocurcumin had a spherical-like shape, with a particle size of 50 ± 5.5 nm.

### 2.4. Animals and Experimental Design

#### 2.4.1. Animals

Thirty-six adult male Wistar rats aged 12–14 weeks and weighing 170–175 g were obtained from the animal facility unit, Faculty of Medicine, Alexandria University, Alexandria, Egypt. Rats were housed at a controlled room temperature (25 °C) with a natural light–dark cycle (12/12 h) for two weeks for acclimation, with diet and water ad libitum throughout the experimental period. All animal procedures and experimental protocols were carried out in accordance with the Alexandria University Institutional Animal Care and Use Committee (IACUC) guidelines (AU14-200922-1-9E). During the experiment, every effort was made to reduce the rats’ suffering.

#### 2.4.2. Experimental Design

The rats were divided into 6 groups (6 rats each). Control group: Healthy rats that received no supplements. Nanocurcumin-treated group: The rats were daily treated orally by gavage with nanocurcumin (15 mg/kg BW). Extracted LPS-treated rats: The rats were treated with single I.P. dose of extracted LPS (5 mg/kg BW). Extracted LPS+nanocurcumin rats: The rats were treated with single I.P. dose of extracted LPS (5 mg/kg BW) plus daily oral nanocurcumin (15 mg/Kg BW). Sigma LPS-treated rats: The rats were treated with single I.P. dose of Sigma LPS (5 mg/kg BW). Sigma LPS+nanocurcumin rats: The rats were treated with single I.P. dose of sigma LPS (5 mg/kg BW) plus daily oral nanocurcumin (15 mg/kg BW). The rats were followed for 7 days after the LPS injection and nanocurcumin supplementations daily via oral gavage (Figure 1).

#### 2.4.3. Sample Collection and Tissue Preparations

At the end of the experimental period, all animals were anesthetized with isoflurane inhalation and sacrificed by cervical dislocation. The brain was rapidly removed, washed with (0.9%) saline solution, and cleaned of adhering matters. The two hemispheres were separated; one hemisphere was fixed in normal formalin for the histological study and the second hemisphere was used to obtain the cerebral cortex and hippocampus. The cerebral cortex and hippocampus were divided into two parts: One was used for total RNA isolation for gene expression analysis, and the second part was homogenized in ice-cold phosphate buffered saline (1:9). The homogenates were centrifuged at 10000× *g* for 20 min at 4 °C and the supernatants were stored at −80 °C for further analysis. 

### 2.5. Assays of Neurotransmitters in Brain Tissues

The tissue levels of acetylcholine (Ach), dopamine (DA), serotonin, and acetylcholine esterase were assayed using specific ELISA kits (Biospes, Chongqing, China), according to the manufacturer’s instructions.

### 2.6. Assay of Malondialdehyde (MDA) as Index of Lipid Peroxidation in Brain Tissues

Malondialdehyde in the whole homogenate of the cerebral cortex and hippocampus were determined according to the method from Draper and Hadley [23]. The sample under test was heated with thiobarbituric acid (TBA) at a low pH. The resulting pink chromogen had a maximal absorbance at 532 nm [23].

### 2.7. Assay of Nitric Oxide End Products (NOx) in Brain Tissues

The nitrite and nitrate (named NOx) concentration was determined by simple Griess reaction [24]. Because nitric oxide (NO) has a short half-life, it is preferable to determine nitrite, the stable product of NO, which may be further oxidized to nitrate [24]. Therefore, the Griess reaction was supplemented with the reduction of nitrate to nitrite by cadmium beads [24]. Briefly, 0.5 g cadmium beads (6–7 beads) was added to a microcentrifuge tube for each sample and for each standard. The beads were activated by washing with 1 mL of each of the following in order: H_2_0, 0.1 M HCl, and 0.1 M NH_4_OH (pH 9.6). The deproteinized samples (by ZnSO_4_ precipitation) were added to the activated Cd beads and incubated at room temperature overnight with agitation. After centrifugation, 100 μL of the supernatants was added into duplicate wells, then 50 μL of sulfanilamide solution was added and shaken briefly. An amount of 50 μL of N-(1-Naphthyl) ethylenediamine dihydrochloride, (NED, 12.5 mM) was added and shaken for 5 min at room temperature. Then, the absorbance values at 540 nm was read in a microtiter plate reader (EZ 40 microplate reader, York, UK).

The standard curve was plotted and the concentrations of the samples were estimated from the curve.

### 2.8. Assay of Glutathione (GSH) in Brain Tissues

The enzymatic method described by Griffith et al. was used to measure the total glutathione and GSSG content [25]. This is a sensitive and specific enzymatic method that depends on the oxidation of GSH by 5,5′-dithiobis-(2-nitrobenzoic acid) (DTNB) to yield GSSG and 5-thio-2-nitrobenzoic acid (TNB) [25]. Oxidized GSSG is reduced enzymatically by the action of glutathione reductase and nicotine amide adenine dinucleotide phosphate (NADPH) to regenerate GSH, which reacts again [25]. The rate of TNB formation is monitored at 412 nm and is proportional to the sum of GSH and GSSG present in the sample [25]. The GSSG content was determined by the same assay as total glutathione, but the reduced glutathione was bound by 2-vinylpyridine [25]. Briefly, aliquots of 0.1 mL of 6.0 mM DTNB, 0.7ml of 0.3 mM NADPH, 0.18mL of distilled water, and 10 µL of either the test sample or standards were mixed and incubated for 15 min at 30 °C. The reaction was initiated by the addition of 10 µL of 50 U/mL glutathione reductase. The rate of formation of TNB was monitored by recording the change in the absorbance at 412 nm per minute (∆A/min) using spectrophotometer Spectro UV-VIS Auto (Labomed inc., Los Angeles, CA, USA). Results were subsequently expressed as nmol glutathione/mg protein by dividing the concentration of glutathione in the sample by the protein concentration in the same sample. The GSSG content was determined by the same assay as total glutathione, but the reduced glutathione is bound by 2-vinylpyridine.

### 2.9. Assays of Tumor Necrosis Factor-α (TNF-α) and Amyloid Beta Protein 1-42 (Aβ1-42) in Brain Tissues

The tissue levels of a TNF-α and Aβ1-42 were assayed using specific ELISA kits (Biospes, Chongqing, China) according to the manufacturer’s instructions. Both kits use the quantitative sandwich enzyme immunoassay technique. Antibody specific for TNF-α or Aβ1-42 was precoated onto a microplate. Standards and samples are pipetted into the wells and any TNF-α or Aβ1-42 present is bound by the immobilized antibody. After removing any unbound substances, a biotin-conjugated antibody specific for TNF-α or Aβ1-42 is added to the wells. After washing, avidin-conjugated horseradish peroxidase (HRP) is added to the wells. Following a wash to remove any unbound avidin-enzyme reagent, a substrate solution is added to the wells and color develops in proportion to the amount of TNF-α or Aβ1-42 bound in the initial step. The color development is stopped, and the intensity of the color is measured using a microplate reader (EZ 40 microplate reader, Biochrome, York, UK).

### 2.10. Assays of Caspase-3 Activity in Brain Tissues

Caspase-3 has a specificity for cleavage at the terminal side of aspartate residue of the amino acid sequence DEVD (Asp-Glu-Val-Asp). The caspase-3 enzymatic activity was assayed using Caspase-3 Assay Kit (Elabscainces, Houston, TX, USA). This kit is used to conjugate caspase-3 sequence-specific peptide acetyl-Asp-Glu-Val-Asp p-nitroanilide (Ac-DEVD-pNA) to yellow group p-nitroaniline (pNA). When the substrate is cut by caspase-3, the yellow group pNA is dissociated. pNA has an absorption peak at 405 nm. The OD value was measured at 405 nm and then the caspase-3 activity was indirectly calculated. The assay was performed in a total volume of 100 µL in a 96-well plate and the absorbance was read using a plate reader. To 50 µL of the tissue homogenate supernatant, 40 µL of caspase-3 assay buffer and 10 µL of Ac-DEVD-*p*NA substrate were added. The plate was covered and incubated at 37 °C for 2 h. The absorbance in the wells was measured at 405 nm using a microplate reader (EZ 40 microplate reader, York, UK). The *p*NA standard curve was constructed by diluting the pNA stock solution (10 mM) in DMSO to prepare serial dilutions between 0 and 100 pmol/µL.

### 2.11. Gene Expression of Brain-Derived Neurotropic Factor (BDNF), Mitochondrial Transcription Factor A (mTFA), β-Site APP Cleaving Enzyme 1 (BACE1), and Nuclear Factor Erythroid 2-Related Factor 2 (Nrf2)

Quantitative analysis of BDNF, mTFA, BACE1, and Nrf2 expression were performed using two-step quantitative real-time reverse transcriptase polymerase chain reaction (qRT-PCR) based on SYBR-Green I fluorescence. First, the total RNA was isolated from the tissues with the RNeasy Mini Kit (Qiagen^®^, Hilden, Germany) and the extracted RNA was reverse-transcribed by QuantiTect Reverse Transcription Kit (Qiagen^®^, Hilden, Germany) following the manufacturer instructions. Then, complementary DNA (cDNA) were amplified and detected using specific primers (Table 1) with real-time PCR. A normalizer or reference gene (18s rRNA) was used as internal control for experimental variability in this type of quantification. The PCR reaction was carried out using the amplification profile as follows: initial denaturation at 95 °C for 10 min, followed by 45 cycles of denaturation (95 °C, 15 s), annealing (55 °C, 15 s), and extension (60 °C, 15 s). The relative quantification of expression was quantified relative to the expression of 18s rRNA using ΔΔCt method or Livak method [26]. Quantitative PCR assay was carried out using Rotor-Gene SYBR Green PCR Kit (Qiagen^®^, Hilden, Germany).

### 2.12. Histological Examination

The hemispheres were fixed with 10% paraformaldehyde and embedded in paraffin to make the brain hemisphere blocks. Then, several serial sections of the blocks were cut at 5 μm thickness using a rotary microtome. After dewaxing the paraffin sections, they were stained with hematoxylin for 3 s, then stained with eosin for 3 min. Following this, the sections were passed through graded alcohols, rinsed in xylene, and sealed. Some paraffin sections were routinely stained with hematoxylin and eosin, as described by Nobakht et al. [27]. The histopathological lesions were investigated under a light microscope and photographed using a digital camera (Nikon Corporation Co., Ltd., Tokyo, Japan).

### 2.13. Statistical Analysis

Data were fed to the computer and analyzed using IBM SPSS software package, version 20.0. (Armonk, NY, USA: IBM Corp). The Shapiro–Wilk test was used to verify the normality of distribution. The data were expressed as mean ± standard deviation (SD). Comparisons between different groups were made using one-way ANOVA followed by Tukey’s post hoc test. Significance of the obtained results was judged at the 5% level.

## 3. Results

### 3.1. Effect of LPS and Nanocurcumin on the Brain Neurotransmitters

The results of the brain neurotransmitters of the studied groups are shown in Table 2.

#### 3.1.1. Acetylcholine (Ach)

In the cortex, the control rats supplemented with nanocurcumin showed a significant increase in the level of Ach compared with the control rats. The rats treated with LPS (Extracted or commercial) showed significantly lower Ach levels than the control rats. The cosupplementation of the LPS-treated rats with nanocurcumin significantly increased the cortical content of Ach compared with unsupplemented LPS-treated rats (extracted or commercial). There was no significant difference between the rats treated with extracted LPS compared with the rats treated with commercial LPS (Table 2).

In the hippocampus, the supplementation of normal rats with nanocurcumin showed no significant effects on the Ach level. The rats treated with LPS (extracted or commercial) showed significantly lower levels of Ach compared with the control rats. The cosupplementation of the LPS-treated rats (extracted or commercial) with nanocurcumin significantly increased the hippocampal content of Ach compared to the unsupplemented LPS-treated rats (extracted or commercial). There was no significant difference between the rats treated with extracted LPS compared with the rats treated with commercial LPS (Table 2).

#### 3.1.2. Acetylcholine Esterase (AchE)

In the cortex, supplementation with nanocurcumin significantly increased the AchE content in the control rats. The rats treated with LPS (extracted or commercial) showed significantly higher content of AchE compared with the control rats or rats supplemented with nanocurcumin. The cosupplementation of LPS-treated rats (extracted or commercial) with nanocurcumin significantly reduced the cortical content of AchE compared with the unsupplemented LPS-treated rats (Table 2).

In the hippocampus, the rats treated with LPS (extracted or commercial) showed significantly higher content of AchE compared with the control rats. Those rats treated with the commercial LPS had significantly higher AchE content than the extracted LPS-treated rats. The cosupplementation of the extracted LPS-treated rats with nanocurcumin significantly normalized the cortical content of AchE, while the supplementation of commercial LPS-treated rats with nanocurcumin significantly reduced the AchE content compared with unsupplemented rats, but was still significantly higher than the other groups (Table 2).

#### 3.1.3. Serotonin

In the cortex, the control rats supplemented with nanocurcumin showed a significant increase in the level of serotonin compared with the control rats. The rats treated with LPS (extracted or commercial) showed significantly lower levels of serotonin than the control rats, with no significant difference between the extracted and commercial LPS. The cosupplementation of LPS-treated rats (extracted or commercial) with nanocurcumin significantly normalized the cortical content of serotonin compared with unsupplemented LPS-treated rats (extracted or commercial) (Table 3).

In the hippocampus, the rats treated with extracted LPS showed mild but significantly lower serotonin levels, while those treated with commercial LPS showed a marked decline in the serotonin levels compared with the control rats. The cosupplementation of LPS-treated rats (extracted or commercial) with nanocurcumin did not significantly affect the hippocampal content of serotonin compared with unsupplemented LPS-treated rats (Table 3).

#### 3.1.4. Dopamine

In the cortex, the control rats supplemented with nanocurcumin showed a nonsignificant decline in the level of dopamine compared with the control rats. The rats treated with LPS (extracted or commercial) showed significantly lower levels of dopamine compared with the control rats. The cosupplementation of LPS-treated rats (extracted or commercial) with nanocurcumin significantly increased the cortical content of dopamine compared with unsupplemented extracted LPS-treated rats and completely normalized in extracted LPS-treated rats (Table 3).

In the hippocampus, no significant difference was observed in the dopamine content between all studied groups (Table 3).

#### 3.1.5. BDNF Expression

In the cortex, the control rats supplemented with nanocurcumin showed a nonsignificant increase in the expression of BDNF compared with the control rats. The LPS-treated rats (extracted or commercial) had significantly lower BDNF expression compared with the control rats. The LPS-treated rats (extracted or commercial) cosupplemented with nanocurcumin showed completely normalized NRF2 expression (Figure 2).

In the hippocampus, the control rats supplemented with nanocurcumin showed no significant change in the BDNF expression compared with the control rats. The rats treated with extracted LPS (cosupplemented or unsupplemented with nanocurcumin) showed no significant change in the BDNF expression compared with the control rats. On the other hand, the rats treated with the commercial LPS showed a significant reduction in BDNF gene expression compared with the control groups. The cosupplementation of the commercial LPS-treated rats with nanocurcumin significantly increased the BDNF expression compared with the unsupplemented rats (Figure 2).

### 3.2. Antioxidant Systems

#### 3.2.1. Glutathione System in Cerebral Cortex

The control rats supplemented with nanocurcumin showed no significant difference in the level of total GSH compared with the control rats. In addition, all other groups of rats showed no significant changes in the levels of total glutathione compared with the control rats (Table 4).

Regarding the level of reduced glutathione (GSH), the rats treated with LPS (extracted or commercial) showed significantly lower cortical GSH levels compared with the control rats. The cosupplementation of LPS-treated rats with nanocurcumin increased the level of GSH compared with the unsupplemented rats; however, these increases are not significant (Table 4).

The rats treated with LPS (extracted or commercial) showed significantly higher levels of GSSG compared with the control rats. The commercial LPS-treated rats had significantly higher GSSG levels compared with the extracted LPS-treated rats. The cosupplementation of LPS-treated rats (extracted or commercial) with nanocurcumin significantly reduced the cortical content of GSSG compared with unsupplemented LPS-treated rats (extracted or commercial). The rats treated with extracted LPS cosupplemented with nanocurcumin had significantly lower GSSG compared with the commercial LPS-treated rats cosupplemented with nanocurcumin (Table 4).

The redox ratio (GSH/GSSG) in the cortex combined the changes in GSH and GSSG. The ratio was significantly higher in the control rats supplemented with nanocurcumin compared with the control rats. The rats treated with LPS (extracted or commercial) had a marked reduction in the GSH/GSSG ratio compared with the control group, with no significant difference between the extracted and commercial LPS. The cosupplementation of the LPS-treated rats (extracted or commercial) with nanocurcumin significantly raised the cortical GSH/GSSG ratio compared with unsupplemented LPS-treated rats (Table 4).

#### 3.2.2. Glutathione System in Hippocampus

The control rats supplemented with nanocurcumin showed no significant difference in the level of GSH compared with the control rats. The rats treated with LPS (extracted or commercial) showed significantly lower total GSH levels compared with the control rats. The cosupplementation of LPS-treated rats does not significantly affect the total GSH levels compared with the unsupplemented rats (Table 5).

The levels of reduced GSH) are significantly lower in rats treated with LPS (extracted or commercial) compared with the control rats with the lowest level detected in the commercial LPS-treated rats. The cosupplementation of LPS-treated rats (extracted or commercial) with nanocurcumin significantly increased the level of GSH compared with the unsupplemented (Table 5).

The rats treated with LPS (Extracted or commercial) showed significantly higher levels of GSSG compared with the control rats, the commercial LPS-treated rats have significantly higher GSSG levels compared with the extracted LPS-treated rats. The cosupplementation of LPS-treated rats (extracted or commercial) with nanocurcumin significantly declined the hippocampal content of GSSG compared with unsupplemented LPS-treated rats. The rats extracted LPS-treated cosupplemented with nanocurcumin have near the normal level and significantly lower GSSG compared with the commercial LPS-treated rats cosupplemented with nanocurcumin (Table 5).

The rats treated with LPS (extracted or commercial) have a marked reduction in the GSH/GSSG ratio compared with the control group with no significant difference between the extracted and commercial LPS. The cosupplementation of LPS-treated rats (extracted or commercial) with nanocurcumin significantly raise the cortical GSH/GSSG ratio compared with unsupplemented LPS-treated rats (Table 5).

#### 3.2.3. Nuclear Factor-Erythroid Factor 2-Related Factor 2 (NRF2) Expression in Cerebral Cortex and Hippocampus

In the cortex, the LPS-treated rats (extracted or commercial) have lower NRF2 expression levels. These effects are not significant compared with the control rats. In addition, the rats treated with the extracted or commercial LPS cosupplemented with nanocurcumin showed significantly higher expressions of NRF2 compared with the unsupplemented rats (Figure 3).

In the hippocampus, the rats treated with extracted LPS showed significant suppression of the NRF2 expression compared with the control rats and the cosupplementation of these rats with nanocurcumin significantly normalized the expression. On the other hand, the rats treated with the commercial LPS showed a significantly marked reduction in the NRF2 gene expression compared with the control groups. The cosupplementation of the commercial LPS-treated rats with nanocurcumin significantly increased the NRF2 expression compared with the unsupplemented rats (Figure 3).

### 3.3. Oxidative Stress Markers

#### 3.3.1. Malondialdehyde (MDA)

The In the cortex, the rats treated with LPS (extracted or commercial) showed significantly higher levels of MDA compared with the control rats. The cosupplementation of LPS-treated rats (extracted or commercial) with nanocurcumin significantly reduced the cortical content of MDA compared with unsupplemented LPS-treated rats (extracted or commercial). There was no significant difference between the rats treated with commercial LPS compared with the rats treated with extracted LPS (Figure 4).

In the hippocampus, the rats treated with LPS (extracted or commercial) showed significantly higher levels of MDA compared with the control rats. The rats treated with commercial LPS had significantly higher MDA levels compared with the rats treated with extracted LPS. The cosupplementation of LPS-treated rats (extracted or commercial) with nanocurcumin significantly reduced the hippocampal content of MDA compared with unsupplemented LPS-treated rats (extracted or commercial) (Figure 4).

#### 3.3.2. Nitric Oxide End Products (NOx)

In the cortex, the LPS-treated rats (extracted or commercial) showed significantly higher levels of NOx compared with the control rats, with the highest level of NOx detected in commercial LPS-treated rats. The cosupplementation of LPS-treated rats (extracted or commercial) with nanocurcumin significantly reduced the cortical content of NOx compared with unsupplemented LPS-treated rats (extracted or commercial). There was a significant increase in levels of NOx in the rats treated with commercial LPS compared with the rats treated with extracted LPS (Figure 5).

In the hippocampus, the rats treated with LPS (extracted or commercial) showed significantly higher levels of NOx compared with the control rats, with the highest level of NOx detected in commercial LPS-treated rats. The cosupplementation of LPS-treated rats (extracted or commercial) with nanocurcumin significantly reduced the hippocampal content of NOx compared with unsupplemented LPS-treated rats (extracted or commercial) (Figure 5).

### 3.4. Markers of Amyloidogenic Pathway

#### 3.4.1. Expression of BACE1

In the cortex, the rats treated with LPS (extracted or commercial) showed significantly higher BACE1 expression compared with the control rats. The cosupplementation of LPS-treated rats (extracted or commercial) with nanocurcumin significantly normalized the cortical expression of BACE1 compared with unsupplemented LPS-treated rats (extracted or commercial). There was no significant difference between the rats treated with extracted LPS compared with the rats treated with commercial LPS (Figure 6).

In the hippocampus, the expression of BACE1 showed no significant changes between the studied groups (Figure 6).

#### 3.4.2. Amyloid Beta Protein 1-42 (Aβ 1-42)

In the cortex, the rats treated with LPS (extracted or commercial) showed significantly higher levels of AB 1-42 compared with the control rats. The supplementation of LPS-treated rats (extracted or commercial) with nanocurcumin significantly reduced the cortical content of AB 1-42 compared with unsupplemented LPS-treated rats (extracted or commercial). There was no significant difference between the rats treated with extracted LPS compared with the rats treated with commercial LPS (Figure 7).

In the hippocampus, the rats treated with LPS (extracted or commercial) showed significantly higher levels of AB 1-42 compared with the control rats. The rats treated with commercial LPS had a significantly higher level of AB 1-42 compared with those treated with the extracted LPS. The supplementation of LPS-treated rats (extracted or commercial) with nanocurcumin did not significantly affect the hippocampal content of AB 1-42 compared with unsupplemented LPS-treated rats (extracted or commercial) (Figure 7).

#### 3.4.3. Marker of Neuroinflammation: TNF-α

In the cortex, the rats treated with LPS (extracted or commercial) showed significantly higher levels of TNF-α compared with the control rats. The commercial LPS-treated rats had significantly higher TNF- α levels compared with the extracted LPS-treated rats. The cosupplementation of LPS-treated rats (extracted or commercial) with nanocurcumin significantly reduced the cortical content of TNF-α compared with unsupplemented LPS-treated rats (extracted or commercial). The rats treated with extracted LPS cosupplemented with nanocurcumin had significantly lower TNF-α compared with the commercial LPS-treated rats cosupplemented with nanocurcumin (Figure 8).

In the hippocampus, the rats treated with extracted LPS showed significantly higher TNF-α levels by about 112%, while those treated with commercial LPS showed significantly higher TNF-α levels by about 417% compared with the control rats. The rats treated with commercial LPS had marked TNF-α levels compared with those treated with the extracted LPS. The supplementation of LPS-treated rats (extracted or commercial) with nanocurcumin caused a significant decline in the hippocampal content of TNF-α compared with unsupplemented LPS-treated rats (extracted or commercial). However, the commercial LPS-treated rats cosupplemented with nanocurcumin still had significantly higher marked TNF-α levels than the other groups (Figure 8).

#### 3.4.4. Caspase-3 Activity aa Apoptotic Marker

In the cortex, the rats treated with LPS (extracted or commercial) showed significantly higher activities of caspase-3 compared with the control rats. The cosupplementation of LPS-treated rats (extracted or commercial) with nanocurcumin significantly reduced the cortical activities of caspase-3 compared with unsupplemented LPS-treated rats (extracted or commercial). There was no significant difference between the rats treated with extracted LPS compared with the rats treated with commercial LPS (Figure 9).

In the hippocampus, the rats treated with extracted LPS (supplemented or unsupplemented with nanocurcumin) showed significantly higher activities of caspase-3 compared with the control rats. Those rats treated with the commercial LPS had significantly higher activity compared with those treated with the extracted one. The cosupplementation of the LPS-treated rats with nanocurcumin showed significantly lower caspase-3 activity compared with the unsupplemented rats (Figure 9).

#### 3.4.5. Expression of Mitochondrial Transcription Factor A (mTFA):

In the cortex, the LPS-treated rats (extracted or commercial) had lower mTFA expression levels. These effects are not significant compared with the control rats. In addition, the rats treated with the extracted or commercial LPS and cosupplemented with nanocurcumin showed no significant effects on the expression of mTFA compared with the unsupplemented rats (Figure 10).

In the hippocampus, the rats treated with extracted LPS showed significant suppression of the mTFA expression compared with the control rats and the cosupplementation of these rats with nanocurcumin showed no significant effect compared with the unsupplemented rats. On the other hand, the rats treated with the commercial LPS showed significantly reduced marked mTFA expression compared with the other groups. The cosupplementation of the commercial LPS-treated rats with nanocurcumin did not significantly affect the mTFA expression compared with the unsupplemented rats (Figure 10).

#### 3.4.6. Histological Examination

At the histological level, the exposure to LPS did not induce any marked changes in the histological structure of the neurons or nerve fibers in the cerebral cortex (Figure 11).

## 4. Discussion

In our lab, we successfully extracted the LPS from *E. coli* ATCC 25922 and examined its neurotoxic effects on the brains of male rats compared with the commercially obtained LPS. Both types of LPS induced similar neurotoxic effects through the induction of histological changes, neurotransmitter disturbances, induction of the amyloidogenic pathway, shifting the redox status, and the induction of inflammation, oxidative stress, and apoptosis in both the cerebral cortex and the hippocampus. Such changes should be associated with a behavioral change; however, we did not perform such behavioral tests in the present study. The understandings of the neurotoxic mechanisms of LPSs are mandatory to identify the most proper methods of protection against LPS neurotoxicity. This study demonstrates the potential protective role of orally administrated nanocurcumin against the LPS-induced neurotoxicity.

At the histological level, the exposure to LPS did not induce marked changes in the histological structure of neurons or nerve fibers. Soudi et al. showed in their preclinical study that, at a dose four-times the dose administered in this study, a moderate infiltration of inflammatory cells in the brain tissue was observed along with one aggregate in the meninges and congestion of the choroid plexus [28].

At the neurotransmitter level, a single I.P. injection of both types of LPS (extracted or commercial) caused a significant decline in the levels of ACh, serotonin, and dopamine in both the cerebral cortex and the hippocampus. The status was worse in the rats treated with the commercial LPS than those treated with the extracted one. These abnormalities are associated with a significant increase in the tissue content of the AchE protein. The cosupplementation of LPS-treated rats with daily oral nanocurcumin significantly ameliorated the levels of neurotransmitters and the AchE protein content in the brain tissues, specifically in the cerebral cortex. Tyagi et al. observed increased AchE activity in their study, along with a marked increase in oxidative stress and an acute inflammatory response in the brain regions linked with several neurodegenerative diseases [29]. In the periphery and the brain, AchE degrades Ach rapidly to regulate its levels [29].

The effect of LPS on the serotonergic system is well established and was demonstrated through an increase in the proinflammatory cytokines, causing anhedonia and a sickness behavior in the rats receiving I.P. LPS [30]. Furthermore, the effects of LPS on the serotonergic system suggests that the LPS might be implicated in an increase in serotonin metabolism and its removal from the synaptic cleft in rat brains [30]. The nanocurcumin cosupplementation significantly corrected the abnormal levels of both serotonin and dopamine in brain tissues. Curcumin proved its efficacy in protecting dopaminergic neurons in rat brains in a Parkinson’s disease model, induced through the administration LPS in a dose-dependent manner [12]. Furthermore, curcumin inhibited the structural altering effect of LPS on rat microglia and further reduced the expressions of nuclear factors kB (NF-kB) and activator protein-1 (AP-1) that were induced by the administration of LPS in vitro [12]. These effects of the orally administered nanocurcumin on Ach, AchE, serotonin, and dopamine point out its possible neuroprotective and anti-inflammatory role, which might aid in limiting the progress of multiple neurodegenerative diseases.

The neurotransmitter changes induced by LPS were associated with a significant decline in the cortex and hippocampus expressions of brain-derived neurotropic factor (BDNF). BDNF plays an important role in the activation of neurogenesis and the suppression of apoptosis, along with the modulation of synaptic activity. Thus, BDNF has an important role in several neurological diseases [31]. Guan et al. previously demonstrated the detrimental effects of LPS on BDNF in the rat hippocampus and cortex [32].

In accordance with our data, suppressed BDNF is associated with neurodegenerative diseases with neuronal loss, such as Alzheimer’s disease (AD), Parkinson’s disease (PD), Huntington’s disease (HD), and bipolar disease [31]. Oral nanocurcumin supplementation proved effective in normalizing the level of BDNF in the rat cerebral cortex and hippocampus.

Another important pathway that is affected by LPS is the amyloidogenic pathway. LPS caused marked induction in the expression of BACE1, which is associated with the marked elevation of Aβ_1-42_. These abnormalities are more prominent in the cerebral cortex than in the hippocampus. Moreover, the commercial LPS was a more powerful inducer of Aβ_1-42_ accumulation than the extracted LPS BACE1 code for β-secretase enzyme, which initiates the cleavage of amyloid precursor protein (APP) at the β-secretase site to produce amyloid beta-protein 1-42 (Aβ_1-42_) [33]. The activated amyloidogenic pathway may indicate the risk for development of AD. Li and colleagues reported that the increase in BACE1 is mediated through an increase in AB_1-42_, demonstrating a high load of AB plaques and an increased risk of developing Alzheimer’s [34]. After nanocurcumin supplementation, the expression of the BACE1 levels in the cortex and hippocampus were completely normalized and the levels of Aβ_1-42_ were significantly reduced in the cortex of rats treated with both types of LPS. In the hippocampus, there was no significant ameliorating effect of the cosupplementation with nanocurcumin.

Regarding neuroinflammation, LPS-injected rats showed marked elevation in the levels of the proinflammatory cytokine, TNF-α, especially in the rats treated with the commercial LPS. The highest elevation in TNF-α was observed in the hippocampus of rats treated with commercial LPS. It has also been shown that TNF-α is involved in enhancing neutrophil recruitment into sites of inflammation. The increase in TNF-α levels after a single I.P. injection of LPS into rats was previously reported by Bossu et al. [5]. Nanocurcumin cosupplementation significantly ameliorated the neuroinflammation by decreasing the elevated TNF-α in the cerebral cortex or hippocampus. Interestingly, curcumin, in its crude form, was reported to decrease proinflammatory cytokines, including TNF-α, in human subjects [35]. These findings further stratify the anti-inflammatory properties of nanocurcumin.

At the redox level, both types of LPS induced significant reductions in the two main antioxidant systems in the brain: The glutathione (GSH) system and the nuclear factor erythroid 2–related factor 2 (NRF2) system. The levels of the reduced GSH were markedly depleted, while the level of the oxidized (GSSG) was significantly elevated, which caused a marked reduction in the redox ratio (GSH/GSSG) in the cerebral cortex and hippocampus of the LPS-treated rats. In addition, the cortex and hippocampus expression of NRF2 was significantly suppressed—especially in the hippocampus. These abnormalities in the redox systems were more prominent with the commercial LPS than the extracted one. Although the increase was not significant in the levels of total and reduced GSH after nanocurcumin supplementation, the level of GSSG significantly decreased, leading to a significant increase in the GSH/GSSG ratio in the cortical tissue. Furthermore, similar results were obtained in the hippocampal tissue. Being essential for the removal of the toxic ROS in the brain, the glutathione system plays a significant role in the protection of the brain tissue from the damage induced by oxidative stress [36]. The impaired glutathione system would render the brain more susceptible to oxidative damage from ROS [36]. Rahman et al. observed similar results on the supplementation of curcumin to D-galactose and normal-ageing mice [37]. In addition, the NRF2 levels in rat brain tissues were significantly rescued by nanocurcumin cosupplementation. This effect was similar across the cortex and hippocampus. Being responsible for the regulation of the basal and induced expression of multiple antioxidant response-element-dependent genes, NRF2 controls the pathophysiological outcomes of the exposure to ROS [38]. Furthermore, the activation of the NRF2 pathway was studied as a potential treatment approach for amyotrophic lateral sclerosis, in which motor neurons progressively degenerate due to their vulnerability to oxidative stress [39].

The observed LPS depletion of the antioxidant systems was associated with marked elevation in the levels of malondialdehydes (MDA), the marker of lipid peroxidation and nitric oxide end products (NOx) in both cortex and hippocampus tissues. Together, impaired antioxidants and increased reactive oxygen species (ROS) production exacerbates the oxidative stress in brain tissues by LPS treatments. The elevated levels of MDA may be due to the overproduction of ROS induced by LPS, which was initially faced by the glutathione, and which results in the depletion of reduced GSH and elevation of GSSG. The disrupted redox status and induced oxidative stress of the brain tissues are associated with the induction of neuroinflammation and apoptosis, as indicated by the marked elevation of TNF-α levels and caspase-3 activity in the brain cortex and hippocampus. The rats receiving cosupplementation with nanocurcumin showed a marked decline in the levels of MDA, NOx, and caspase-3 activity. These effects are similar across the cortex and hippocampus. Rajasekar and colleagues obtained similar results with MDA in rats after supplementation with nanocurcumin crystals [40]. In addition, the reduction in NOx would probably be related to the anti-inflammatory effect of curcumin. Furthermore, orally administered curcumin nanoemulsions showed similar results on TBARS and NOx in rats [41].

In addition to being an apoptotic marker, caspase-3 has an important role in the process of normal brain development through the formation of apoptotic bodies [42]. Nanocurcumin supplementation has led to a marked decline in the levels of caspase-3, further protecting the brain tissue from apoptosis. This effect was observed only in the cortex but not the hippocampus of rat brains.

At the molecular level, LPS significantly suppressed the gene expression of mitochondrial transcription factor A (mTFA), a transcriptional factor controlling the mitochondrial biogenesis in brain tissues through maintaining the copy number and structure of the mitochondrial DNA (mtDNA), as well as the transcription and replication of the mtDNA [43]. Mitochondria are the powerhouses of the cell, as they generate most of the ATP required by the cell. The brain is a highly metabolic organ and neurons in the central nervous system have an intense demand for mitochondria [44]. As such, the suppressed expression of mtTFA in the brain tissues of rats treated with LPS may indicate a decreased mitochondrial biogenesis and mitochondrial DNA replication and transcription that may lead to mitochondrial dysfunction and neuronal damage.

The mechanism by which LPSs suppress the brain gene expression of mtTFA is unclear. However, we can postulate that LPSs may have the ability to pass through the BBB to reach neuron cells, causing a direct consequence of oxidative stress and increased free radicals and lipid peroxide, which is confirmed in this study. The cosupplementation of LPS-treated rats with nanocurcumin improved the brain expression of the mTFA gene, which may completely normalize the levels in the cortex and hippocampus of the extracted LPS-treated rats and the cortex only in commercial-LPS treated rats, which may imply restoring mitochondrial biogenesis and function.

Collectively, the extracted LPS and the commercially obtained LPS (Sigma Aldrich) showed nearly similar effects on most of the studied parameters, aside from some instances where the commercially obtained LPS produced a more pronounced effect. This difference in outcome between the two sources of LPS could be related to a difference in the species from which the LPS was isolated, since the O-antigen segment is different in the LPS extracted from any bacteria, even within the same species [45].

In general, nanocurcumin particles showed neuroprotective and antioxidant effects, rescuing rats injected intraperitoneally with LPS endotoxin.

## 5. Conclusions

LPSs have potential neurotoxic effects that significantly impact brain neurotransmitters, antioxidant systems, and mitochondrial biogenesis, and elicit neuroinflammation and apoptosis, which may increase the risk of neurological diseases. The study confirmed the protective effects of oral nanocurcumin against the LPS-induced neurotoxicity in the cerebral cortex and hippocampus, and recommends the application of nanocurcumin for the protection and/or treatment against septicemia-induced neurotoxicity.

## Figures and Tables

**Figure 1 biomedicines-10-03087-f001:**
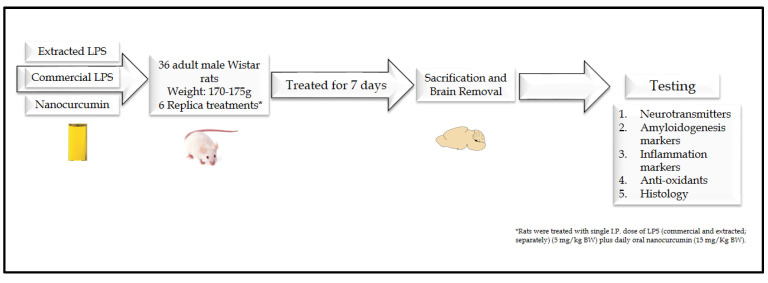
Experimental design.

**Figure 2 biomedicines-10-03087-f002:**
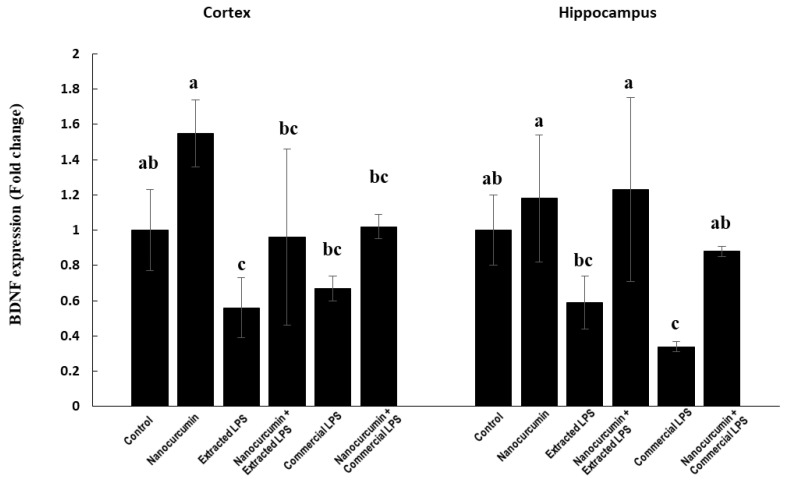
BDNF in control rats and rats treated with nanocurcumin and/or LPS (extracted or commercial). Data are expressed using mean ± SD. Six replicas for each group. Groups were compared at *p* < 0.05 using one-way ANOVA and Tukey’s post hoc test. Those which are not assigned with a shared letter (a, b, and c) are statistically significant. E-LPS: Extracted lipopolysaccharides, C-LPS: Commercial lipopolysaccharides.

**Figure 3 biomedicines-10-03087-f003:**
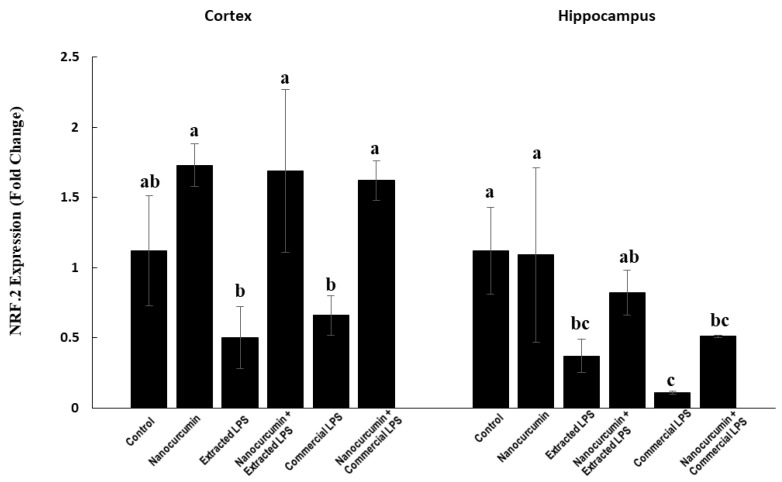
NRF2 in control rats and rats treated with nanocurcumin and/or LPS (extracted or commercial). Data are expressed using mean ± SD. Six replica for each group. Groups were compared at *p* < 0.05 using one-way ANOVA and Tukey post hoc test. Those which are not assigned with a shared letter (a, b, and c) are statistically significant. E-LPS: Extracted lipopolysaccharides, C-LPS: Commercial lipopolysaccharides.

**Figure 4 biomedicines-10-03087-f004:**
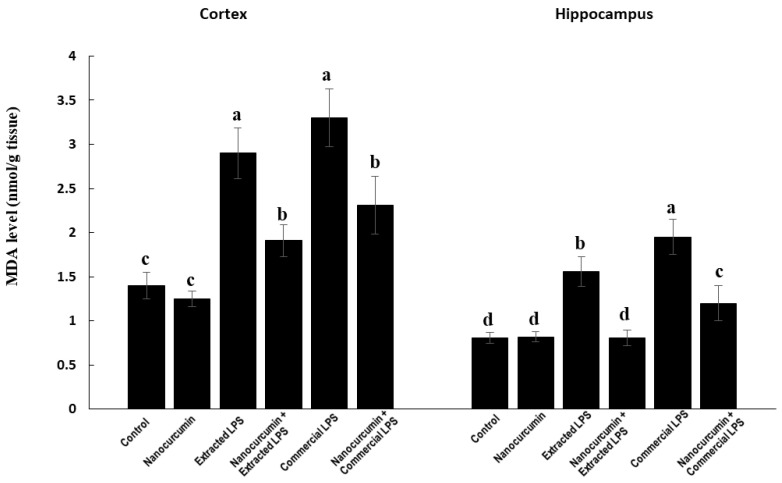
MDA in control rats and rats treated with nanocurcumin and/or LPS (extracted or commercial). Data are expressed using mean ± SD. Six replica for each group. Groups were compared at *p* < 0.05 using one-way ANOVA and Tukey’s post hoc test. Those which are not assigned with a shared letter (a, b, c, and d) are statistically significant. E-LPS: Extracted lipopolysaccharides, C-LPS: Commercial lipopolysaccharides.

**Figure 5 biomedicines-10-03087-f005:**
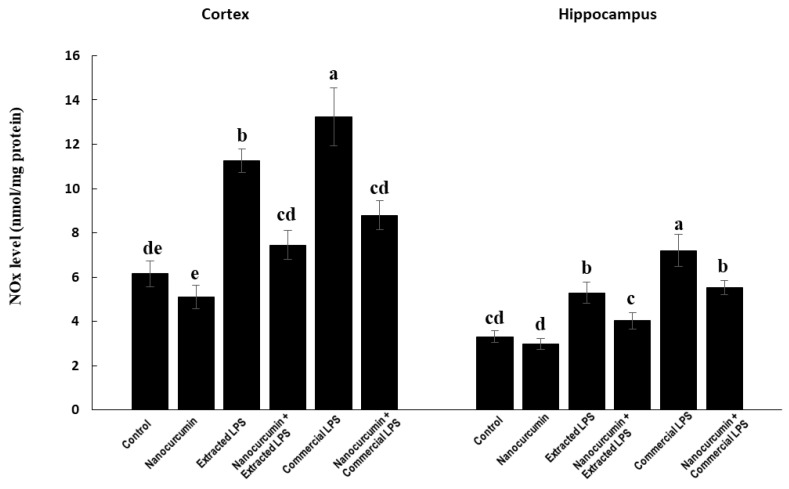
NOx in control rats and rats treated with nanocurcumin and/or LPS (extracted or commercial). Data are expressed using mean ± SD. Six replica for each group. Groups were compared at *p* < 0.05 using one-way ANOVA and Tukey’s post hoc test. Those which are not assigned with a shared letter (a, b, c, d, and e) are statistically significant. E-LPS: Extracted lipopolysaccharides, C-LPS: Commercial lipopolysaccharides.

**Figure 6 biomedicines-10-03087-f006:**
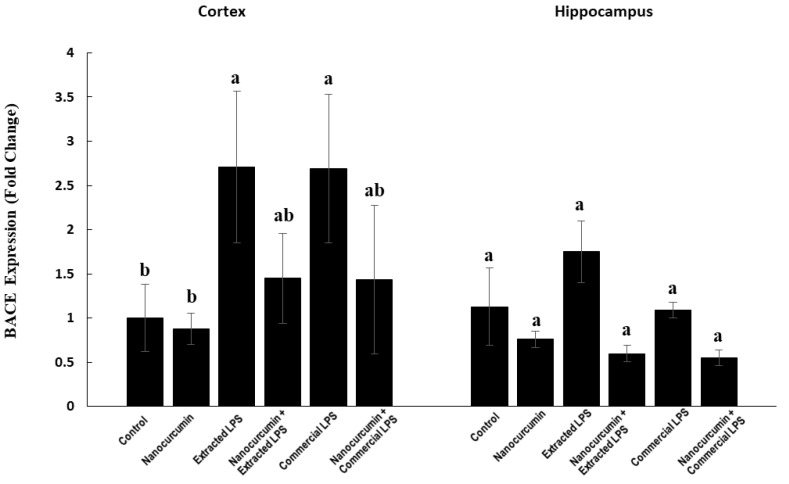
BACE in control rats and rats treated with nanocurcumin and/or LPS (extracted or commercial). Data are expressed using mean ± SD. Six replicas for each group. Groups were compared at *p* < 0.05 using one-way ANOVA and Tukey’s post hoc test. Those which are not assigned with a shared letter (a and b) are statistically significant. E-LPS: Extracted lipopolysaccharides, C-LPS: Commercial lipopolysaccharides.

**Figure 7 biomedicines-10-03087-f007:**
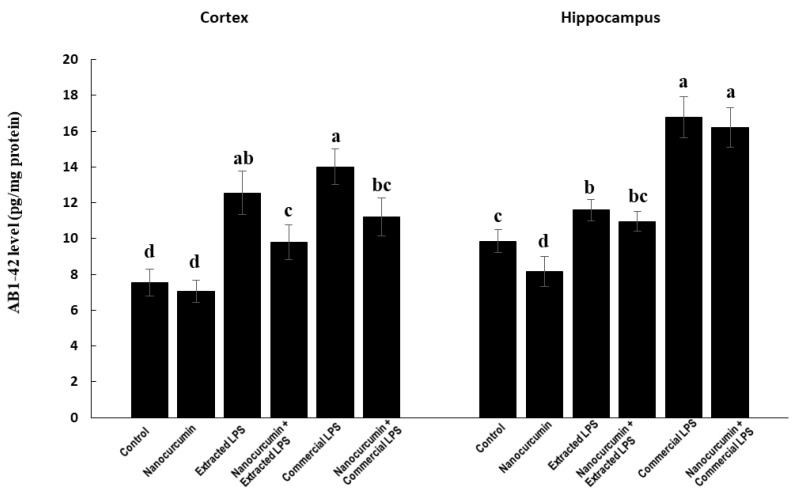
AB1-42 in control rats and rats treated with nanocurcumin and/or LPS (extracted or commercial). Data are expressed using mean ± SD. Six replicas for each group. Groups were compared at *p* < 0.05 using one-way ANOVA and Tukey post hoc test. Those which are not assigned with a shared letter (a, b, c, and d) are statistically significant. E-LPS: Extracted lipopolysaccharides, C-LPS: Commercial lipopolysaccharides.

**Figure 8 biomedicines-10-03087-f008:**
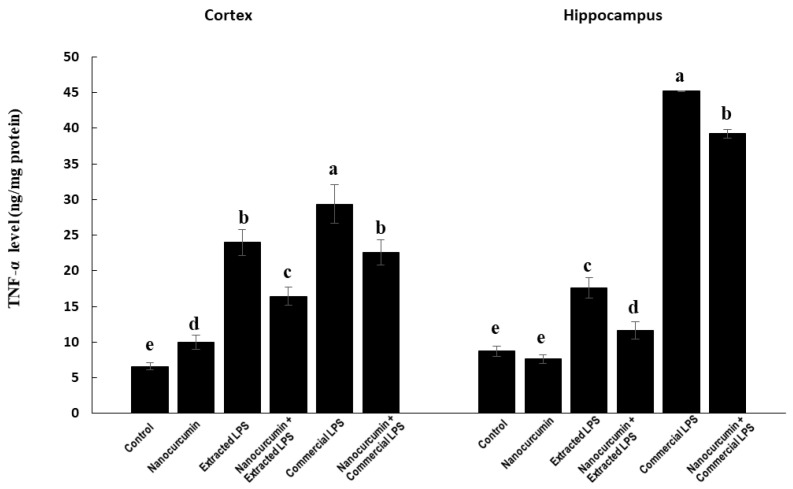
TNF-A in control rats and rats treated with nanocurcumin and/or LPS (extracted or commercial). Data are expressed using mean ± SD. Six replicas for each group. Groups were compared at *p* < 0.05 using one-way ANOVA and Tukey’s post hoc test. Those which are not assigned with a shared letter (a, b, c, d, and e) are statistically significant. E-LPS: Extracted lipopolysaccharides, C-LPS: Commercial lipopolysaccharides.

**Figure 9 biomedicines-10-03087-f009:**
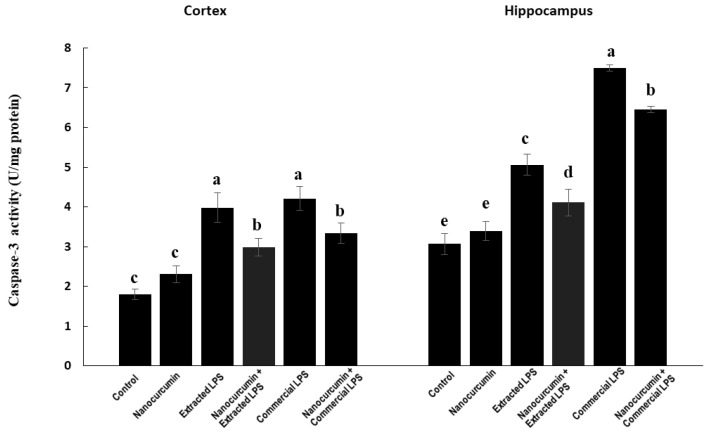
Caspase-3 in control rats and rats treated with nanocurcumin and/or LPS (extracted or commercial). Data are expressed using mean ± SD. Six replicas for each group. Groups were compared at *p* < 0.05 using one-way ANOVA and Tukey’s post hoc test. Those which are not assigned with a shared letter (a, b, c, d, and e) are statistically significant. E-LPS: Extracted lipopolysaccharides, C-LPS: Commercial lipopolysaccharides.

**Figure 10 biomedicines-10-03087-f010:**
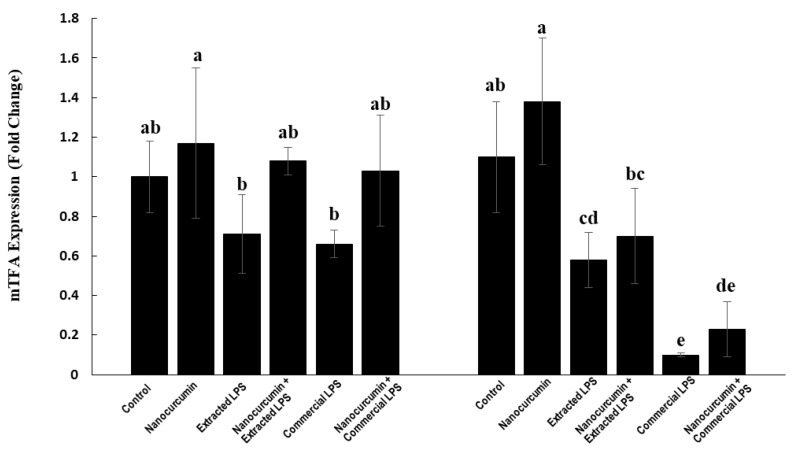
mTFA in control rats and rats treated with nanocurcumin and/or LPS (extracted or commercial). Data are expressed using mean ± SD. Six replicas for each group. Groups were compared at *p* < 0.05 using one-way ANOVA and Tukey’s post hoc test. Those which are not assigned with a shared letter (a, b, c, d, and e) are statistically significant. E-LPS: Extracted lipopolysaccharides, C-LPS: Commercial lipopolysaccharides.

**Figure 11 biomedicines-10-03087-f011:**
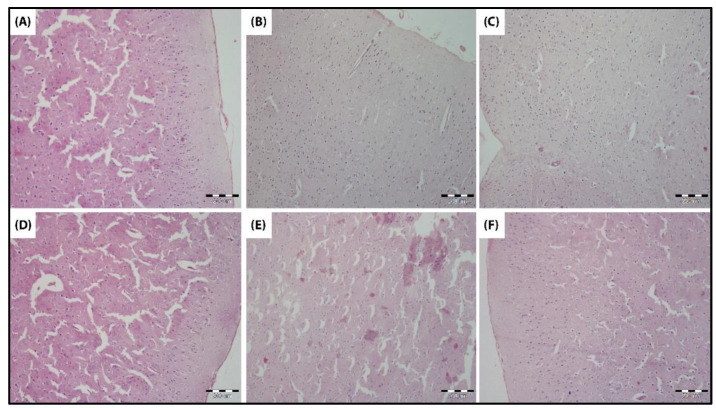
Histological examination of the cerebral cortex of different groups of male Wistar rats (hematoxylin-eosin staining at 200 µm). (**A**) Control, (**B**) nanocurcumin, (**C**) extracted LPS, (**D**) nanocurcumin + extracted LPS, (**E**) commercial LPS, and (**F**) nanocurcumin + commercial LPS.

**Table 1 biomedicines-10-03087-t001:** Primer sequences of rat BACE1, BDNF, Nrf2, and mTFA, and 18s rRNA.

Gene	Accession No.	Primer Sequence	
BACE1	NM_019204.2	F	GCATGATCATTGGTGGTATC
		R	CCATCTTGAGATCTTGACCA
BDNF	NM_001270638.1	F	GAGAAGAGTGATGACCATCCT
		R	TCACGTGCTCAAAAGTGTCAG
NRF2	NM_031789.2	F	CAAATCCCACCTTGAACACA
		R	CGACTGACTAATGGCAGCAG
mTFA	NM_031326.2	F	CCCTGGAAGCTTTCAGATACG
		R	AATTGCAGCCATGTGGAGG
18s rRNA	NR_046237.2	F	GTAACCCGTTGAACCCCATT
		R	CAAGCTTATGACCCGCACTT

**Table 2 biomedicines-10-03087-t002:** The cortex and hippocampus content of acetylcholine (Ach) and acetylcholine esterase (AchE) in control rats and rats treated with nanocurcumin and/or LPS (extracted or commercial).

	Ach (ng/mg Protein)		AchE (ng/mg Protein)	
	Cortex	Hippocampus	Cortex	Hippocampus
**Control**	6.41 ^b^ ± 0.23	17.26 ^a^ ± 1.52	11.73 ^d^ ± 0.90	20.11 ^d^ ± 1.52
**Nanocurcumin**	7.46 ^a^ ± 0.53	16.97 ^a^ ± 1.35	15.37 ^c^ ± 0.90	20.79 ^d^ ± 1.72
**E-LPS**	4.19 ^c^ ± 0.29	11.21 ^b^ ± 1.15	22.46 ^ab^ ± 2.08	28.91 ^c^ ± 2.30
**E-LPS + Nanocurcumin**	6.33 ^b^ ± 0.62	15.79 ^a^ ± 1.33	15.97 ^c^ ± 1.20	23.24 ^d^ ± 2.02
**C-LPS**	4.40 ^c^ ± 0.18	10.73 ^b^ ± 0.08	24.86 ^a^ ± 2.26	43.61 ^a^ ± 0.08
**C-LPS + Nanocurcumin**	6.13 ^b^ ± 0.45	15.31 ^a^ ± 0.33	19.49 ^b^ ± 1.74	37.95 ^b^ ± 1.05

Data are expressed using Mean ± SD. Six replica for each group. Groups were compared at *p* < 0.05 using one-way ANOVA and Tukey post hoc test. Those which are not assigned with a shared letter (a, b, c, and d) are statistically significant. E-LPS: Extracted lipopolysaccharides, C-LPS: Commercial lipopolysaccharides.

**Table 3 biomedicines-10-03087-t003:** The cortex and hippocampus content of serotonin and dopamine in control rats and rats treated with nanocurcumin and/or LPS (extracted or commercial).

	Serotonin (ng/mg Protein)		Dopamine (ng/mg Protein)	
	Cortex	Hippocampus	Cortex	Hippocampus
**Control**	3.85 ^b^ ± 0.42	4.96 ^a^ ± 0.52	5.73 ^a^ ± 0.44	10.27 ^a^ ± 0.73
**Nanocurcumin**	4.49 ^a^ ± 0.33	4.92 ^ab^ ± 0.43	5.06 ^ab^ ± 0.36	10.66 ^a^ ± 0.98
**E-LPS**	2.50 ^d^ ± 0.23	4.25 ^b^ ± 0.31	4.20 ^c^ ± 0.38	10.41 ^a^ ± 0.98
**E-LPS + Nanocurcumin**	3.37 ^bc^ ± 0.24	4.41 ^ab^ ± 0.39	5.18 ^ab^ ± 0.44	10.24 ^a^ ± 0.72
**C-LPS**	3.01 ^cd^ ± 0.23	2.82 ^c^ ± 0.08	4.13 ^c^ ± 0.38	10.44 ^a^ ± 0.08
**C-LPS + Nanocurcumin**	3.87 ^ab^ ± 0.39	2.97 ^c^ ± 0.25	4.84 ^bc^ ± 0.47	10.27 ^a^ ± 0.62

Data are expressed using mean ± SD. Six replica for each group. Groups were compared at *p* < 0.05 using one-way ANOVA and Tukey’s post hoc test. Those which are not assigned with a shared letter (a, b, c, and d) are statistically significant. E-LPS: Extracted lipopolysaccharides, C-LPS: Commercial lipopolysaccharides.

**Table 4 biomedicines-10-03087-t004:** The glutathione system in the cerebral cortex of control rats and rats treated with nanocurcumin and/or LPS (extracted or commercial).

	Total GSH (nmol/mg Protein)	Reduced GSH (nmol/mg Protein)	GSSG (nmol/mg Protein)	GSH/GSSG Ratio
**Control**	4.79 ^ab^ ± 0.48	4.22 ^a^ ± 0.44	0.28 ^cd^ ± 0.03	14.93 ^b^ ± 1.44
**Nanocurcumin**	5.11 ^a^ ± 0.42	4.64 ^a^ ± 0.40	0.24 ^d^ ± 0.03	19.95 ^a^ ± 2.58
**E-LPS**	4.07 ^b^ ± 0.30	3.06 ^b^ ± 0.24	0.51 ^b^ ± 0.04	6.04 ^d^ ± 0.54
**E-LPS + Nanocurcumin**	4.14 ^b^ ± 0.32	3.46 ^b^ ± 0.30	0.34 ^c^ ± 0.03	10.14 ^c^ ± 1.23
**C-LPS**	4.17 ^b^ ± 0.42	2.95 ^b^ ± 0.30	0.61 ^a^ ± 0.06	4.84 ^d^ ± 0.48
**C-LPS + Nanocurcumin**	4.24 ^b^ ± 0.42	3.35 ^b^ ± 0.30	0.49 ^b^ ± 0.02	8.93 ^c^ ± 0.49

Data are expressed using mean ± SD. Six replicas for each group. Groups were compared at *p* < 0.05 using one-way ANOVA and Tukey post hoc test and those which are not assigned with a shared letter (a, b, c, and d) are statistically significant. E-LPS: Extracted lipopolysaccharides, C-LPS: Commercial lipopolysaccharides.

**Table 5 biomedicines-10-03087-t005:** The glutathione system in the hippocampus of control rats and rats treated with nanocurcumin and/or LPS (extracted or commercial).

	Total GSH (nmol/mg Protein)	Reduced GSH (nmol/mg Protein)	GSSG (nmol/mg Protein)	GSH/GSSG Ratio
**Control**	5.98 ^a^ ± 0.58	5.27 ^ab^ ± 0.59	0.36 ^c^ ± 0.04	15.04 ^a^ ± 2.87
**Nanocurcumin**	6.24 ^a^ ± 0.66	5.58 ^a^ ± 0.59	0.33 ^c^ ± 0.04	17.04 ^a^ ± 1.62
**E-LPS**	4.68 ^bc^ ± 0.40	3.53 ^c^ ± 0.33	0.57 ^b^ ± 0.04	6.15 ^cd^ ± 0.54
**E-LPS + Nanocurcumin**	5.30 ^ab^ ± 0.48	4.50 ^b^ ± 0.44	0.40 ^c^ ± 0.04	11.19 ^b^ ± 1.26
**C-LPS**	3.87 ^c^ ± 0.39	2.43 ^d^ ± 0.24	0.72 ^a^ ± 0.07	3.38 ^d^ ± 0.34
**C-LPS + Nanocurcumin**	4.49 ^bc^ ± 0.39	3.39 ^c^ ± 0.25	0.55 ^b^ ± 0.07	8.42 ^bc^ ± 0.34

Data are expressed using Mean ± SD. Six replicas for each group. Groups were compared at *p* < 0.05 using one-way ANOVA and Tukey’s post hoc test. Those which are not assigned with a shared letter (a, b, c, and d) are statistically significant. E-LPS: Extracted lipopolysaccharides, C-LPS: Commercial lipopolysaccharides.

## Data Availability

The data generated in this study are available within the article.

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
