# Peer review of "The Neuroprotective and Antioxidant Effects of Nanocurcumin Oral Suspension against Lipopolysaccharide-Induced Cortical Neurotoxicity in Rats"

_biomedicines, 2022, doi:10.3390/biomedicines10123087_

Round 1

Reviewer 1 Report

The submitted manuscript titled with “The protective effects of nano-curcumin against lipopolysaccharide-induced neurotoxicity in rats” has done some experiments to explore the pathways involved in curcumin’ protection roles in vivo. Overall, the novelty of this manuscript is not enough since such topic has been well-addressed in previous investigations and the authors did not provide novel pathways.
Major concerns:
1) Since commercial LPS is available, why the authors would like to produce LPS by themselves? The rational is not there and there was not much meaning to compare these two LPS from difference resources.
2) What is the meaning of nano-curcumin? What is the difference and advantage by using nano-curcumin? Does this one can reach higher concentration in the brain comparing to other curcumin?
3) The authors should compare different curcumin but not different LPS since the authors would like to demonstrate the protective effects of this novel format curcumin.
4) The reviewer is also confused on figure legend “Means in the column with common superscript letters are not significant and Means with Different superscript letters are significant by ANOVA followed by Tukey post hoc test; p<0.05”. The description should more accurate and clearer.
5) The whole manuscript is mainly a descriptive investigation not a mechanism study.

Author Response

The submitted manuscript titled with “The protective effects of nano-curcumin against lipopolysaccharide-induced neurotoxicity in rats” has done some experiments to explore the pathways involved in curcumin’ protection roles in vivo.  Overall, the novelty of this manuscript is not enough since such topic has been well-addressed in previous investigations and the authors did not provide novel pathways.

Major concerns:

  • Since commercial LPS is available, why the authors would like to produce LPS by themselves? The rational is not there and there was not much meaning to compare these two LPS from difference resources.

Response:  Due to the high cost of the commercial LPS (Ex Sigma LPS 5mg cost about 300 $) so we have a project for local production of LPS and the present study was a part of this project. This method is pending Egyptian patent, so to evaluate its biological activity, the extracted LPS was compared with standard LPS of Sigma Aldrich. Based on our results, we are going to use the extracted LPS in our future work due to the expensive price of Sigma LPS.

  • What is the meaning of nano-curcumin? What is the difference and advantage by using nano-curcumin? Does this one can reach higher concentration in the brain comparing to other curcumin?

Response:  the introduction section was rewritten to illustrate the difference and advantage of using nan0-curcumin.  Nano-curcumin can cross the blood–brain barrier (BBB) to enter brain tissue, where it was concentrated chiefly in the hippocampus and significantly prolonged retention time in the cerebral cortex (increased by 96%) and hippocampus (increased by 83%). (Ref: Tsai, Y. M., Chien, C. F., Lin, L. C., & Tsai, T. H. (2011). Curcumin and its nano-formulation: the kinetics of tissue distribution and blood-brain barrier penetration. International journal of pharmaceutics, 416(1), 331–338. https://doi.org/10.1016/j.ijpharm.2011.06.030)

  • The authors should compare different curcumin but not different LPS since the authors would like to demonstrate the protective effects of this novel format curcumin.

Response:  the difference between the curcumin as native or as a nanoformulation has been already studied and the superior effect of nanofromulation over the native form is well documented (Ex. Dende C, Meena J, Nagarajan P, Nagaraj VA, Panda AK, Padmanaban G. Nanocurcumin is superior to native curcumin in preventing degenerative changes in Experimental Cerebral Malaria. Sci Rep. 2017 Aug 30;7(1):10062. doi: 10.1038/s41598-017-10672-9. PMID: 28855623; PMCID: PMC5577147.)

  • The reviewer is also confused on figure legend “Means in the column with common superscript letters are not significant and Means with Different superscript letters are significant by ANOVA followed by Tukey post hoc test; p<0.05”. The description should more accurate and clearer. 

Response:  The figures and Tables legends were corrected in the revised manuscript to be clearer.

  • The whole manuscript is mainly a descriptive investigation not a mechanism study.

Response:  off course part of our study is descriptive; however, the rest is mechanistic as we explored different mechanisms of LPS-induced neurotoxicity and compared different types of LPS. Also, we make intervention with nanocurcumin to explore the possible protective effects against LPS-induced neurotoxicity.

Reviewer 2 Report

Paper titled (The protective effects of nano-curcumin against lipopolysaccharide-induced neurotoxicity in rats) by Salah et al. discussed the beneficial effect of a nanopreparation of curcumin in protection from the LPS model of neurotoxicty induced in male rats. The main problem here is in Introduction & some parts of the methods are not described adequately. Also, authors need to explore the novelty more.

1- Title: mention the type of nanopreparation

2- Title: mention the mechanism by which nanocurcumin produced the protective effect

3- Introduction: Authors wrote too much on LPS, While did not introduce curcumin or the value ofnanopreparation & which type of nanotechnologies.

4- Methods: give an account on type of the purchased nanoparticles & if some data about charcaters. What is the type of nanotechnology used in synthesis,

5- What was the value of using 2 types of LPS?

6- What was the type and code number of ELISA kits & procedures.

7- Authors should give the source of chemicals, kits and antibodies completely and consistently (code, company, town, state and country) & version for software

8 - Details of reading and used instruments should be mentioned (ex: NO assay)

9- What was the method & reference used for quantification of PCR results

10- Mention method of assessment of HE stained sections in Methods.

11- This reviewer cannot find the HE results or figure!!

12- Some figures can be combined to reduce the total number

13- SD value is large in some groups although authors confirmed the data are in normal distribution.

14- In cocnlusion, write the clinical impact of this study.

Author Response

12- Some figures can be combined to reduce the total number

Response: Unfortunately, we cannot combine the figures anymore

13- SD value is large in some groups although authors confirmed the data are in normal distribution.

Response: The high SD values are only present in the gene expression assays; BACE1, mTFA, and NRF2 which is usually found in such assay, however the results are still normally distributed according to the Shapiro-Wilk test as indicated in the section of statistical analysis.

14- In conclusion, write the clinical impact of this study.

Response: Done

Reviewer 3 Report

Review for biomedicines-1972329

            The manuscript by Adham Salah et al., entitled “The protective effects of nano-curcumin against lipopolysaccharide-induced neurotoxicity in rats” is a research article that reported the findings that Nanocurcumin administration enhances antioxidants and neuroprotection against the LPS-induced neurotoxicity in rats. Authors use different techniques to confirm the outcomes. The study was well designed by the authors and the outcomes are significant. The manuscript is well written, and the cited references are appropriate. However, some minor remarks should be taken by authors under consideration before paper publication. The manuscript needs minor revision before its final publication.

 Comments:

1.     In the materials and methods section, the authors should include the age of the rats used for this study.

2.     In the materials and methods section, the authors should add the experimental design as a figure. It will be better understandable for the treatment timeline.

3.     In the statistical section, the authors should add detailed statistical methods used for the study. For example, the authors should explain the One-way ANOVA method and what post-hoc method was used for the significance like details.

4.     Authors demonstrated changes in neurotransmitter levels in the cortex and hippocampus of experimental rats. Changes in neurotransmitter levels must affect the behavior in rats. So, the authors might analyze the behavior changes in the experimental rats. It will give more supporting evidence for the research outcome.

5.     In figures 1-9, authors should include the group labeling on the “X” axis and place the brain region name on the top of the graph. It will be well served to the readers.

6.     Authors mentioned that no histological changes were observed in the cortex and hippocampus. The authors should represent the histopathological images of the cortex and hippocampus.

Author Response

Reviewer 3 comments

 The manuscript by Adham Salah et al., entitled “The protective effects of nano-curcumin against lipopolysaccharide-induced neurotoxicity in rats” is a research article that reported the findings that Nanocurcumin administration enhances antioxidants and neuroprotection against the LPS-induced neurotoxicity in rats. Authors use different techniques to confirm the outcomes. The study was well designed by the authors and the outcomes are significant. The manuscript is well written, and the cited references are appropriate. However, some minor remarks should be taken by authors under consideration before paper publication. The manuscript needs minor revision before its final publication.

 Comments:

  1. In the materials and methods section, the authors should include the age of the rats used for this study.

Response: the age of rats was 12-14 weeks as indicated in the revised manuscript.

  1. In the materials and methods section, the authors should add the experimental design as a figure. It will be better understandable for the treatment timeline.

Response: This comment have been resolved by adding the experimental design as a figure.

  1. In the statistical section, the authors should add detailed statistical methods used for the study. For example, the authors should explain the One-way ANOVA method and what post-hoc method was used for the significance like details.

Response: The data were expressed as mean ± standard deviation (SD). Comparisons between different groups were made using one-way ANOVA followed by Tukey post-hoc test.

  1. Authors demonstrated changes in neurotransmitter levels in the cortex and hippocampus of experimental rats. Changes in neurotransmitter levels must affect the behavior in rats. So, the authors might analyze the behavior changes in the experimental rats. It will give more supporting evidence for the research outcome.

Response: Thank you for this valuable suggestion and we realized the importance of such behavioral test but unfortunately all animals sacrificed, and we cannot perform such behavioral test now. So, we documented this test as a limitation of our study (lines: 587-588).

  1. In figures 1-9, authors should include the group labelling on the “X” axis and place the brain region name on the top of the graph. It will be well served to the readers.

Response: This comment have been resolved by amending the figures.

  1. Authors mentioned that no histological changes were observed in the cortex and hippocampus. The authors should represent the histopathological images of the cortex and hippocampus.

Response: a figure of histological analysis of H and E stain were added in the revised manuscript.

Round 2

Reviewer 1 Report

Although the authors did some responses to the reviewer's comments, the overall quality of this manuscript has not been increased significantly.  The rational for using two types of LPS is not solid.      

Author Response

Although the authors did some responses to the reviewer's comments, the overall quality of this manuscript has not been increased significantly.  The rational for using two types of LPS is not solid.

Response: Dear Reviewer, thank you for your previous comments, suggestions, and recommendations. We would like to elaborate more on the rational for using two types of LPS in our study,

Due to the high cost of research materials in Egypt, the national direction is to produce many biochemical research materials locally. The commercial LPS of which 5mg costs about 300 $ (Equivalent to 6000 EGP), which in turn exerts a financial burden over researchers in Egypt.

LPS is one of many toxins used in our lab in the institute of graduate studies and research in Alexandria, Egypt. We were able to secure funds through a project for local production of LPS and the present study was part of this project.

The LPS in our lab was extracted by hot phenol-water method as described in (El-Moslemany, R. M.; El-Kamel, A. H.; Allam, E. A.; Khalifa, H. M.; Hussein, A.; Ashour, A. A. Tanshinone IIA loaded bioactive nanoemulsion for alleviation of lipopolysaccharide induced acute lung injury via inhibition of endothelial glycocalyx shedding. Biomed. Pharmacother. 2022, 155, 113666; DOI:10.1016/j.biopha.2022.113666.) With some modifications.

The previously mentioned method is pending Egyptian patent, so to evaluate the biological activity of the extracted LPS it was compared with standard LPS of Sigma Aldrich. We adopted this approach based on some published articles in which the authors used the extracted lipopolysaccharide and compared it with that of Sigma Aldrich to assess the biological and immunological activities of their extracted lipopolysaccharides such as;

  • Correa, W., Brandenburg, K., Zähringer, U., Ravuri, K., Khan, T., & Von Wintzingerode, F. (2017). Biophysical analysis of lipopolysaccharide formulations for an understanding of the low endotoxin recovery (LER) phenomenon. International journal of molecular sciences18(12), 2737.‏
  • Kianmehr, Z., Ardestani, S. K., Soleimanjahi, H., Fotouhi, F., Alamian, S., & Ahmadian, S. (2015). Comparison of biological and immunological characterization of lipopolysaccharides from Brucella abortus RB51 and S19. Jundishapur journal of microbiology8(11).‏

Moreover, evidence in literature indicate that LPS biological activity might differ according to the source as summarized in this review:

  1. Schultz, C. (2018). Lipopolysaccharide, structure and biological effects.  Intern. Med. Clin. Innov3(1.10), 15761.‏

Although the main objective of our study was to describe the neuro-protective effect of orally administered nano-curcumin suspension against LPS-induced neurotoxicity, we also had a secondary objective to evaluate the biological efficacy of our laboratory-extracted LPS in the same settings.

We would like to hear more about your recommendations to enhance the way our rational is presented within the article.

Reviewer 2 Report

The revised form of the paper titled (The protective effects of nano-curcumin against lipopolysaccharide-induced neurotoxicity in rats) was revised in a good manner however, some parts were missed, kindly find the following recommendations:

1- Title: better to say (cortical neurotoxicity)

2- Fig 11: quality of images is not high, need high resolution images & quantification of the neurons in the cortex by image analysis used by an experienced pathologist

3- Curcumin nanoparticles :Authors give account in introduction OK, but did not inform in methods about the type used in their hands !!! give details about the technique used for this preparation

Author Response

The revised form of the paper titled (The protective effects of nano-curcumin against lipopolysaccharide-induced neurotoxicity in rats) was revised in a good manner however, some parts were missed, kindly find the following recommendations:

  • Title: better to say (cortical neurotoxicity)

Response:  The title of the paper was amended to be “The neuro-protective and anti-oxidant effects of nano-curcumin oral suspension against lipopolysaccharide-induced cortical neurotoxicity in rats”

  • Fig 11: quality of images is not high, need high resolution images & quantification of the neurons in the cortex by image analysis used by an experienced pathologist

Response:  the images were added with higher quality. Thank you for such constructive recommendations to perform neuron quantification using image analysis. However, due to the fact that we finalized our experiment 8 months ago, we were unable to repeat the histological examination as we no longer have residual tissue to carry out the histological examination and obtain higher magnification and more images to pursue such analysis.

We would like to hear your feedback whether we should remove the image or leave it as it is.

  • Curcumin nanoparticles: Authors give account in introduction OK, but did not inform in methods about the type used in their hands!!! give details about the technique used for this preparation

Response: as per the company of origin of the purchased nanocurcumin, the nanocurcumin powder was prepared by emulsification-solvent evaporation using Annaraj et al method (Annaraj, J.; Dhivya, R.; Vigneshwar, M.; Dharaniyambigai, K.; Kumaresan, G.; Rajasekaran, M. J. NanoSci. NanoTech, 2014, 2, 490. ISSN: 2279-0381).

Bulk curcumin was dissolved in chloroform in the ratio of 1:2 (solution A), PVA dissolved in distilled water (solution B). Drop wise addition of Solution A to boiling water at 70°C under stirring and continued stirring for further one hour. At the end of the reaction, an orange colored fluffy mass was obtained. Solution B was added into solution A and the temperature was reduced below 40°C, the stirring was continued for another 2 hours. The solution was allowed to settle and stored at below -14°C for one day. The supernatant was discarded after centrifugation and the reaction solution was dried in a water bath at 60°C, a fine powder of PVA loaded nanocurcumin was obtained.

PVA loaded nanocurcumin thus obtained was characterized by UV-Vis absorption spectra that was obtained on an Ocean Optics USB2000+VIS-NIR Fiber optics spectrophotometer. Furthermore to check the shape and size, TEM was performed on JEOL JEM-2100 high resolution transmission electron microscope at an accelerating voltage of 200 kV. The obtained nanocurcumin had a spherical like-shape with a particle size of 50 ± 5.5 nm.
